# Basalt Fiber Hybridization Effects on High-Performance Sisal-Reinforced Biocomposites

**DOI:** 10.3390/polym14071457

**Published:** 2022-04-03

**Authors:** Bernardo Zuccarello, Francesco Bongiorno, Carmelo Militello

**Affiliations:** Dipartimento di Ingegneria, Università degli Studi di Palermo, 90128 Palermo, Italy; francesco.bongiorno01@unipa.it (F.B.); carmelo.militello01@unipa.it (C.M.)

**Keywords:** biocomposites, natural fibers, sisal, basalt, aging, mechanical performance

## Abstract

The increasing attention given to environmental protection, largely through specific regulations on environmental impact and the recycling of materials, has led to a considerable interest of researchers in biocomposites, materials consisting of bio-based or green polymer matrixes reinforced by natural fibers. Among the various reinforcing natural fibers, sisal fibers are particularly promising for their good mechanical properties, low specific weight and wide availability on the current market. As proven in literature by various authors, the hybridization of biocomposites by synthetical fibers or different natural fibers can lead to an interesting improvement of the mechanical properties or, in turn, of the strength against environmental agents. Consequently, this can lead to a significant enlargement of their practical applications, in particular from quite common non-structural applications (dashboards, fillings, soundproofing, etc.) towards semi-structural (panels, etc.) and structural applications (structural elements of civil construction and/or machine components). Hybridizations with natural fibers or with ecofriendly basalt fibers are the most interesting ones, since they permit the improvement of the biocomposite’s performance without an appreciable increment on environmental impact, as occurs instead for synthetic fiber hybridizations that are also widely proposed in the literature. In order to further increase the mechanical performance and, above all, to reduce the aging effects on high-performance sisal-reinforced biocomposites due to environmental agents, the hybridization of such biocomposites with basalt fibers are studied with tensile, compression and delamination tests performed by varying the exposition to environmental agents. In brief, the experimental analysis has shown that hybridization can lead to further enhancements of mechanical performance (strength and stiffness) that increase with basalt volume fraction and can lead to appreciable reductions in the aging effects on mechanical performance by simple hybridization of the surface laminae. Therefore, such a hybridization can be advantageously used in all practical outdoor applications in which high-performance sisal biocomposites can be exposed to significant environmental agents (temperature, humidity, UV).

## 1. Introduction

The use of biocomposites consisting of eco-friendly matrices reinforced by natural fibers has increased rapidly in many industrial sectors such as the automotive, shipbuilding, and civil construction industries. In particular, the use of natural fibers characterized by good mechanical properties, low specific weight, and wide availability in the current market, as is the case of sisal fiber, not only allows compliance with the recent and increasingly stringent regulations of environmental protection, but also has the benefit of an appreciable reduction in production and operating costs, especially for applications in the transport sector (automotive, nautical, etc.) [1,2]. 

Several studies reported in the literature have shown that, generally, the hybridization of these biocomposites with synthetic fibers or different natural fibers, can allow the improvement of their performance, and therefore the expansion of this field of application, by passing from classical non-structural applications to semi-structural and structural, or from indoor applications to outdoor applications, thanks improving their strength against aging from environmental agents (temperature, humidity, UV).

In [3], hybrid epoxy composite laminates based on basalt fiber composites as the inner core, and using glass, flax and hemp fiber laminate to obtain symmetrical configurations, with a 21–23% fiber volume fraction, are considered. Through tensile, three-point bending and interlaminar shear strength tests, whose fracture surfaces have been characterized by scanning electron microscopy, the authors show that the mechanical performance of all the hybrid laminates appear superior to pure hemp and flax fiber-reinforced laminates, and inferior to basalt fiber laminates. Among the hybrids, the best properties were exhibited by those obtained by adding glass and flax to basalt fiber-reinforced laminates. 

An appreciable and rather complete review work is presented in [4], considering both hybrids with natural and synthetic fibers and hybrids with all synthetic fibers. Both thermoplastic and thermoset composites reinforced by natural/synthetic fibers or synthetic/synthetic fibers are considered. The work constitutes an interesting compendium on the subject for the reader, and reference should be made to it for detailed information.

In [5], the author studies the effect of silica microparticle inclusions and the stacking sequence of glass fiber cross-ply fabric and short sisal fiber layers on the apparent density, tensile and flexural strength and modulus of hybrid epoxy composites. It is, however, noteworthy that the incorporation of silica particles improves the mechanical performance of composites containing larger amounts of sisal fibers.

In [6], carbon fiber and flax fiber were used to manufacture different hybridized composites with varying flax fiber volume fractions. Resulting composites were characterized using tensile, flexural, impact, and vibration tests. Moreover, the results of the experiments were compared against the predicted rule of mixtures model and the Halpin–Tsai model. The findings of this study provide valuable information for designers relative to hybridizing flax and carbon fibers, and the results suggest that hybridizing synthetic fibers with natural fibers is an effective method of improving mechanical properties and controlling vibration damping.

In [7], an evaluation of the effect of a layering sequence and chemical treatment on the mechanical properties of hybrid woven kenaf–Kevlar composites was performed. In detail, to evaluate the effect of the chemical treatment, the woven kenaf mat was treated with 6% sodium hydroxide (NaOH)-diluted solution, and the mechanical properties were considered with untreated kenaf hybrid composites. Tensile and flexural properties of treated hybrid composites are better than non-treated hybrid composites. This study is a part of the exploration of the potential application of hybrid composites in high-velocity impact applications.

In [8], the authors present a review work intending to focus on suitable techniques to manufacture bio-sourced hybrid composites. Some of the fabrication methods are customized in order to suit the application of natural fibers. The selected methods are also highlighted for application in the aerospace and automotive industries. The process and outcomes are presented comparatively.

In [9], a review work on the mechanical properties of a composite reinforced with glass fiber alternating with natural fibers (such as abaca, banana, bamboo, cotton, coconut, hemp, jute, pineapple, sisal, etc.) is presented. In particular, it is shown how hybridization with natural fibers can generally improve the mechanical characteristics of these materials.

The review article [10] intended to present information about different classes of natural fibers, nanofiller, cellulosic fiber-based composites, nanocomposites, and natural fiber/nanofiller-based hybrid composites with specific concern to their applications. Moreover, it provides a summary of the emerging new aspects of nanotechnology for the development of hybrid composites for a sustainable and greener environment.

The research work [11] focuses on the development of flax–basalt carbon fiber-reinforced epoxy/bioepoxy hybrid composites under different stacking sequence techniques. A water absorption study under distilled water conditions, contact angle measurements, thermogravimetric analyses, and a surface morphology study using scanning electron microscopy analysis were also performed. From the results obtained, the composites manufactured can be prospectively utilized in various engineering medium-load structural applications.

The main aim of paper [12] was to develop a kenaf–glass fiber-reinforced unsaturated polyester hybrid composite as a source of green composite using a sheet molding compound process. The kenaf fiber was treated with 6% sodium hydroxide (NaOH)-diluted solution for 3 h using the mercerization method. The hybrid composites were tested for flexural, tensile and Izod impact strength. Scanning electron microscopy fractography showed fiber cracking, debonding, and fibers being pulled-out as the main fracture modes of the composites, and the kenaf-treated reinforced hybrid composite exhibited better interfacial bonding between the matrix and the reinforcement compared to other combinations.

In [13], resistance of bamboo fiber-reinforced polypropylene composite (BFRP) and bamboo-glass fiber-reinforced polypropylene hybrid composite (BGRP) to hygrothermal aging and their fatigue behavior under cyclic tensile load were studied. The results suggested that BGRP had better fatigue resistance than BFRP at all load levels tested.

In [14], the authors considered the mechanical properties and the life cycle assessment (LCA) of jute-woven fabric composites and their hybrids. Jute-woven fabric composites were sandwiched with glass-woven composites with the epoxy matrix. The water absorption test was performed on jute-woven composites and composite sandwiches. It showed that thin layers of glass-woven composites in the composite sandwich decelerated water penetration to the jute-woven composites, which were the core materials. A commercial LCA GaBi^®^ software product was employed to evaluate the environmental impacts of manufacturing the jute-woven composites and their hybrids.

The paper [15] presented on the tensile and flexural properties of jute–glass–carbon fiber-reinforced epoxy hybrid composites in an inter-ply configuration. The results showed that the hybridization process can potentially improve the tensile and flexural properties of jute-reinforced composite. The flexural strength decreases when partial laminas from a carbon/epoxy laminate are replaced by glass/epoxy or jute/epoxy laminas. Moreover, it is realized that incorporating high-strength fibers to the outer layers of the composite leads to higher flexural resistance, whilst the order of the layers does not affect the tensile properties.

The review article [16] presented fiber hybridization in polymer composites. The aim was to explain basic mechanisms of these hybrid effects and describe the state-of-the-art models that predict them. An overview of the tensile, flexural, impact, and fatigue properties of hybrid composites was presented to aid in the optimal design of hybrid composites. Finally, some current trends in fiber hybridization, such as pseudo-ductility, were described.

The study [17] investigated carbon (CFRP), jute (NFRP) and hybrid (HFRP) fiber-reinforced polymers manufactured using the resin-transfer molding process. The hybridization of low-cost, sustainable jute with carbon fiber offers a more sustainable and economic alternative to CFRPs, with excellent damping properties.

In [18], an experimental study on tensile, flexural and impact properties of flax–basalt–glass-reinforced epoxy hybrid composites is shown. The effects of reinforcement hybridization, fiber relative amounts, and stacking sequence on the mechanical properties were investigated. The results showed that the developed hybrid composites displayed enhanced tensile, flexural, and impact performance, as compared with flax-reinforced epoxy composite. The mechanical properties of hybrid laminates were proven to be highly dependent on the position of the flax layers within the hybrid composite. The hybridization with basalt and/or glass fibers was found to be an effective method for enhancing the mechanical properties of flax/epoxy composites. 

In summary, it appears relevant to consider that the literature has so far shown that the hybridization of biocomposites with glass or basalt fibers generally makes it possible to improve the mechanical performance and/or to reduce the effects of aging due to environmental agents. This allows for more widespread use, not only in non-structural indoor applications, as is still the case especially in the automotive field (dashboards, fills, etc.), but also in new semi-structural and structural outdoor applications characterized by significant effects related to environmental agents. However, no work in the literature has been devoted to the study of basalt–sisal hybrid biocomposites reinforced by long fibers, and no aging effects due to environmental agents have been considered. In order to evaluate the possibility of obtaining further improvements of the mechanical properties and, above all, significant enhancements in resistance to aging due to environmental agents (temperature, moisture, and UV) of high-performance sisal-reinforced biocomposites (materials that can be used to replace metals and fiber glass in semi-structural and structural applications), the effects of the hybridization obtained through the use of basalt fibers have been analyzed in the present work. This study was essentially performed through tensile, compressive, and delamination tests, carried out on sisal/basalt hybrid biocomposites by varying the exposure to the environmental agents, simulated through accelerated aging tests performed in accordance with the ASTM G 154 standard.

## 2. Materials and Manufacture of Biocomposites

The manufacturing of the hybrid basalt–sisal biocomposites (B-SFRP) was carried out by using twill-type woven fabrics of basalt fibers with a weight of 220 g/m^2^ (supplied by Delta PREG, see Figure 1a), and unidirectional *stitched* fabrics of sisal fibers with a weight of 180 g/m^2^ (properly made in the laboratory). In detail, the sisal fabrics, that are not disposable in the current market, were obtained by manual stretching of the fibers in order to eliminate natural undulations; alignment of the straight fibers arranged into small groups, the transversal stitching by using an automatic machine (see Figure 1b). These unidirectional fabrics, also used to manufacture the pure sisal cross-ply biocomposites (SFRP) considered to compare the performance of the B-FSRP, were the same as those already used in several previous studies by the same authors [19,20,21,22,23,24,25,26,27].

In particular, a batch of optimized sisal fibers with density *ρ_f_* = 1.45 g/cm^3^, tensile strength *σ_f_*_,*R*_ = 685 MPa, Young modulus *E_f_* = 40 GPa, and tensile strain *ε_f_*_,*R*_ = 1.75%, were used for the manufacturing of the sisal fabrics (see also Table 1).

The simple comparison of the above-reported properties of the sisal fiber with those of the basalt fibers, provided by the manufacturer and summarized in Table 1, show how basalt fibers have, in practice, specific weight and Young’s modulus about double that of sisal fibers, whereas tensile strength is about four times higher (2800 MPa against 685 MPa).

The matrix used for both biocomposites considered was a green epoxy produced by the American Entropy Resin Inc. (San Antonio, CA, USA), named SUPERSAP CNR, with IHN-type hardener [28]. As widely shown in previous studies [24,25,26,27] by the same authors, this matrix exhibits an almost linear elastic behavior with density *ρ_m_* = 1.05 g/cm^3^, tensile strength *σ_m_*_,*R*_ 50 MPa, Young modulus *E_m_* ≅ 2.7 GPa, and tensile fracture strain *ε_m_*_,*R*_ ≅ 2.5% (see also Table 1).

Taking into account the specific weight of the fabrics (sisal and basalt), a cross-ply hybrid laminate with the same volume fraction of basalt and sisal, equal to 25%, has been obtained by the stacking sequence [(0/90)_4B_/0_S_/90_S_/(0/90)_4B_/0_S_/90_S_]_S_, for a total thickness of 4 mm. This configuration corresponds to a laminate constituted by 24 laminae, with an overall fiber volume fraction *V_f_* = 50% and density equal to 1.6 g/cm^3^. This laminate represents, in practice, the maximum hybridization, since sisal and basalt fibers have the same volume fraction, and it has been considered to evaluate the maximum effects that the basalt hybridization can give in terms of both mechanical strength and resistance to aging due to environmental agents.

In order to compare the performance of the hybrid biocomposite implemented, a pure sisal biocomposite laminate (SFRP) was manufactured with the same lay-up and same fiber volume fraction (density equal to 1.25 g/cm^3^). In detail, the manufacture of both biocomposites was carried out by hand lay-up, and performed through a plane mould of 260 mm × 260 mm. To obtain high-quality laminates (without voids and/or internal defects) with *V_f_* = 50%, after an appropriate time of partial gelification, the laminate was subjected to an optimized compression-molding process [22] by using a hydraulic press of 100 tons (see Figure 2a). The compression-molding process, which lasted about 5 h, was followed by a post-curing process at 80 °C for 120 min, carried out through the use of an appropriate electrical oven (see Figure 2b).

Figure 3 shows the images of the sisal/basalt hybrid laminate (see Figure 3a,b) and the pure sisal laminate (see Figure 3c,d). In particular, the lateral view of the hybrid biocomposite (Figure 3a) allows the observation of the succession of basalt (dark color) and sisal (light color) laminae.

From these panels, all the specimens necessary for the tensile, compressive and delamination tests were obtained by a proper disk cutter. In detail, for each material and for each test, 5 different specimens were used.

## 3. Accelerated Aging and Experimental Tests

In order to reproduce the effects of aging due to environmental agents (temperature, sunlight, humidity, and UV) that can occur in a common “outdoor” application, the biocomposites produced were subjected to an accelerated aging process with controlled temperature, humidity, and UV radiation conditions. The analysis of the influence of aging on the main mechanical properties of the material were performed by tensile, compressive, and delamination tests.

### 3.1. Accelerated Aging

In more detail, the biocomposite specimens were exposed to UV radiation (UVA-340) and water condensation in an accelerated aging chamber QUV (see Figure 4a); alternating cycles of UVA radiation (8 h) at 70 °C and water condensation (4 h) at 50 °C, for an overall period of 42 days (1008 h) were used. Such an accelerated aging process was carried out according to all the conditions prescribed by the ASTM G154 (Standard Practice for Operating Fluorescent Light Apparatus for UV Exposure of Nonmetallic Materials) [29].

For both the biocomposites considered, three rectangular specimens with dimensions of 80 × 160 mm were placed on a special specimen rack (see Figure 4b) placed inside the aging chamber. Every 336 h, a partial sampling of each material (hybrid and non-hybrid) was performed in order to carry out the tensile, compressive, and delamination tests, for a total of three samples at 336, 672, and 1008 h.

### 3.2. Material Testing

Tensile, compressive and delamination tests were carried out on both the biocomposites by considering the four conditions of virgin materials and aging materials after 336, 772, and 1008 h of accelerated aging. An MTS 793 servo-hydraulic machine was used for all mechanical characterization tests, (tensile, compressive and delamination), by using a proper test device.

In detail, the tensile tests were carried out in accordance with the ASTM D3039 standard [30], by using specimens of size 25 mm × 160 mm, instrumented by an MTS extensometer having a measuring base of 25 mm (see Figure 5a).

The compressive tests were performed in accordance with the ASTM D6641 standard [31], by using similar specimens of 25 mm × 160 mm instrumented by a VISHAY electrical resistance strain gauge (with a measuring base b = 12 mm)—see Figure 5b.

Finally, the interlaminar shear strength ILSS was evaluated in accordance with the ASTM D2344 standard [32] that considered a three-point bending test on a short-beam with a length equal to six times the thickness, i.e., about 24 mm, and width equal to two times the thickness, i.e., about 8 mm. The short-beam test specimens were placed on two side supports, with a span equal to four times the thickness (16 mm)—see Figure 5c.

All tests were carried out with a traverse speed of 1 mm/min and, for each material and for each aging condition considered, three specimens were tested.

## 4. Results

In order to highlight the contribution of fibers on the biocomposite aging, Figure 6 shows the images of the SFRP cross-ply specimens after the different steps of accelerated aging were performed, along with the images of the same aging on a specimen made by neat epoxy. It is to be noted how the accelerated aging process leads to an appreciable variation of color, which goes towards darker and darker shades, from the initial light-yellow color of the virgin material to the dark brown of the fully aged material. In more detail, the comparison of the images of the neat epoxy and the SFRP biocomposite show that this darkening was higher for the biocomposite (especially for high-exposure times), i.e., the fiber aging led to an appreciable contribution to the aging process observed. 

In Figure 7, the images of the cross-ply hybrid samples (B-SFRP) are similarly reported, after the different steps of aging. The visual examination shows, in this case, lower effects of superficial darkening that seem to involve only the matrix, as no appreciable variations in color of the basalt fibers were observed (it visibly retains its characteristic black color).

### 4.1. Tensile Tests

Figure 8 shows the results of the tensile test (average curves), in terms of specific stress index *I_σ_* (ratio between the actual stress and the specific weight of the biocomposite) for the B-SFRP hybrid biocomposite and SFRP reinforced by sisal fiber only. From Figure 8, it is possible to observe how the hybrid biocomposite has a higher failure strain and a superior specific strength, equal to about 20 MPa/(kN/m^3^), appreciably higher than that of the simple SFRP, equal to about 15 MPa/(kN/m^3^). Therefore, hybridization with basalt led to an increase in the specific tensile strength of about 33%.

Considering the material stiffness, represented by the Young modulus, which as we can see from Figure 8 moves from the value of about 0.87 GPa/(kN/m^3^) for the SFRP to the value of about 1.12 GPa/(kN/m^3^) for B-SFRP, it can be said that the hybridization considered gave rise to an improvement in the stiffness index with an increase of 25%, approximately.

Therefore, it can be stated that, for the biocomposites examined, the hybridization with basalt gave the expected improvements in the mechanical performance (+33% and +25% for tensile strength and tensile modulus, respectively), as already described in the literature for other hybrid biocomposites reinforced by basalt fiber.

In Figure 9, the tensile curves of both the SFRP and B-SFRP are compared with the curves obtained for the same biocomposite after 336 h, 672 h, and 1008 h of accelerated aging.

From Figure 9a it is possible to observe that the aging produces an appreciable progressive reduction in the specific tensile strength of the SFRP. On the other hand, from Figure 9b it is possible to observe a much more modest progressive reduction in tensile strength for the hybrid B-SFRP.

From Figure 10 it can be observed that, in practice, the SFRP biocomposite underwent a progressive reduction in tensile strength with an asymptotic behavior corresponding to a relative reduction of about 35%. Instead, the hybridization with basalt significantly limited the effects of aging which, in practice, led to a reduction in tensile performance of less than 13%.

Similarly, in Figure 11, the specific tensile stiffness of the two biocomposites examined (SFRP and B-SFRP) is reported as a function of the accelerated aging hours.

From Figure 11 it can be seen that the biocomposite SFRP underwent a modest reduction in specific tensile stiffness by the accelerated aging, with a maximum relative decrement of about 20%. The hybridization with basalt limited the effects of aging, resulting in a maximum reduction of about 12%.

In order to observe the damage mechanisms governing the failure of the hybrid biocomposite compared to those that ordinarily occur in biocomposites reinforced with sisal fibers only, in Figure 12 and Figure 13 the images of the relative specimens after the tensile test are reported.

In detail, the visual examination of the SFRP specimen after the tensile test (see Figure 12) shows how the failure of the biocomposite involves fiber tensile failure preceded by limited interlaminar delamination phenomena. In more detail, in accordance with the tensile curves shown above, it can be observed that the progressive damage first involved the transverse failure of the laminae at 90°, with limited progressive reduction in the stiffness of the specimen (reduction in the slope of the tensile curve), with appreciable load bridging phenomena (progressive variation of stiffness) and subsequent failure of the fibers of the longitudinal laminae, preceded by limited debonding phenomena accompanied by interlaminar delamination phenomena.

From the examination of Figure 13, it can be observed instead that the failure of the hybrid biocomposite comprised the phenomena of delamination accompanied by the phenomena of fiber–matrix debonding, with evident swelling of the specimen in the useful area between the grips, as occurs typically for hybrid composites with fibers that have a considerable mismatch of elastic characteristics. The extensive delamination phenomena prevent, in this case, the beneficial effects of load *bridging,* and are responsible, together with the different failure strains of the different fibers, for the relatively low performance of the hybrid composite. The replacement of 50% of sisal fibers with basalt fibers having about 4 times greater strength, this should in fact lead to at least a doubling of the tensile strength of B-SFRP, not to a relatively modest increase of 40%, as observed above.

### 4.2. Compressive Tests

The results of the compressive tests for the two biocomposites SFRP and B-SFRP examined before and after accelerated aging for 336, 672, and 1008 h are shown in Figure 14 below.

From the examination of Figure 14 it is possible to observe that the hybridization led to an increment of specific compressive strength of about 110%, from approximately 3.5 MPa/(kN/m^3^) of the SFRP to approximately 7.4 MPa/(kN/m^3^) of the B-SFRP. This result was mainly due to the higher stiffness of the basalt fiber with respect to the sisal fiber, also limiting the transversal strain responsible for the compressive failure. 

In addition, it can be observed that the effects of aging on compressive strength were almost similar for both hybrid and non-hybrid biocomposites. In practice, aging gives rise for both materials to a relative reduction in compressive strength, tending asymptotically to about −1.5 MPa/(kN/m^3^). This result is justified by the fact that, under compressive load, the failure of both biocomposites in practice is due to matrix failure under transversal traction (as shown below in Figure 15, illustrating the damage of the specimens after the compressive test). Consequently, the aging of the sisal fibers had no significant effect on the compressive strength; only the effect of aging on the matrix had significant results, which were clearly identical for both biocomposites. 

However, in relative terms, there were different values of compressive strength reduction due to the higher strength of the hybrid biocomposite, with about 7 MPa/(kN/m^3^) against about 3 MPa/(kN/m^3^) of the biocomposite reinforced by only sisal fibers. In detail, the B-SFRP hybrid biocomposite exhibited a maximum reduction of 18% against about 33% of the SFRP.

From the examination of Figure 15, which refers to the typical hybrid specimen after the compressive test, it can be observed in detail how the compressive failure is characterized by debonding phenomena and intralaminar delamination that initially propagate on the transverse sisal laminae. This is followed by interlaminar delamination that leads to a consequent decrease in stiffness and shear mode instability phenomena against sisal and basalt fibers, with consequent debonding due to the local shear stress.

### 4.3. ILSS Tests

Figure 16 represents the average curves related to the two not-aged biocomposites obtained from ILSS tests performed on short-beam specimens subject to bending at three-points in accordance with ASTM D2344.

In accordance with the considerable mismatch of the elastic properties of the two different fibers, sisal and basalt, as well as with what has already been observed in terms of the damage mechanisms of biocomposites under tensile stress, from Figure 16 it can be seen that the hybrid biocomposite had a lower delamination strength. In other words, in the absence of aging (virgin material), hybridization resulted in a 30% reduction with respect to the performance of the SFRP biocomposite.

This condition is reversed for aging biocomposites, as shown in Figure 17 that represents the delamination strength of the two biocomposites for various aging. This confirms the expected significant beneficial effects produced by hybridization in the presence of significant aging effects from environmental agents.

In quantitative terms, Figure 17 shows how, for the SFRP biocomposite, the aging led to a significant reduction in the ILSS strength with an asymptotic value of about −70%, whereas hybridization significantly limited such negative effects by fully recovering the effects of fiber stiffness mismatch observed before for non-aging biocomposites. The hybrid biocomposites exhibited a percentage delamination strength reduction not exceeding −30%.

Finally, in Figure 18 the images of typical specimens of SFRP and B-SFRP after aging at 1008 h are reported. Qualitatively similar conditions have been detected for partial aging.

The comparative examination of Figure 18a,b shows how aging led to a more marked delamination phenomena on the SFRP specimens, essentially affecting the upper semi-beam subjected to compression, and highlighting a probable (negative) interaction between the effects of shear and compressive stresses. Such effects were not observed in the stiffest hybrid specimen B-SFRP, and that largely justifies the best absolute performance of the latter in the presence of aging.

## 5. Conclusions

In conclusion, therefore, it is possible to state that the experimental study carried out on the effects of the basalt hybridization of high-performance biocomposites reinforced by sisal fibers (SFRP) permitted, in general, an improvement in the mechanical performance of these biocomposites, particularly in the presence of aging due to environmental agents, as occurs in outdoor structural applications.

In detail, the comparison of the tensile, compressive and shear performance of SFRP and hybrid B-SFRP cross-ply laminates—obtained in practice from SFRP laminates by replacing 50% of the laminae with twill-type basalt fabrics—showed that such hybridization results in the following main effects:Appreciable increase in tensile strength of about 33% associated with an increase in tensile stiffness of about 25%;The benefits of hybridization on tensile strength and stiffness increased further with aging. In detail, the hybrid biocomposite B-SFRP exhibited a maximum tensile strength reduction (after 1008 h of accelerated aging) of just −13%, whereas the biocomposite SFRP exhibited a maximum tensile strength reduction of about −35%. In terms of stiffness, it went from a maximum reduction of −20% for the SFRP to a maximum reduction of −12% for the B-SFRP;Moreover, the compressive strength of the SFRP biocomposite underwent significant benefits from hybridization, with noticeable increases of about 110%. In the presence of aging, instead, fewer effects were observed. The hybrid biocomposite allowed, in practice, only a modest decrease in compressive strength reduction due to aging, which in practice passed from 33% of SFRP to 18% of B-SFRP;Finally, unlike tensile and compressive strength, the delamination strength of the SFRP biocomposites did not increase by basalt hybridization, but underwent a reduction of about −30%. However, in the presence of aging, this condition was reversed, and hybridization contributed to significantly limiting the effects of environmental degradation: from a delamination strength reduction of −70% of the SFRP under 1008 h of accelerated aging, to the maximum delamination strength reduction of about −30% of the hybrid B-SFRP.

Further developments of the present work are in progress, and consider in detail sisal-reinforced biocomposites whose upper and bottom laminae are constituted by basalt lamina obtained with a light fabric. Such particular configuration is considered in order to obtain a good enhancement of the aging strength, without the negative effects in terms of delamination strength. The quality of lightweight is also considered, taking into account that basalt fibers contribute to increasing the specific weight of high-performance sisal-reinforced biocomposites.

## Figures and Tables

**Figure 1 polymers-14-01457-f001:**
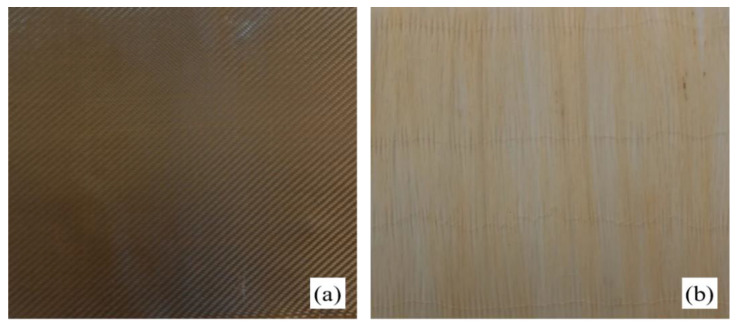
(**a**) Twill-type commercial basalt fabrics and, (**b**) unidirectional sisal-stitched fabrics properly made in the laboratory.

**Figure 2 polymers-14-01457-f002:**
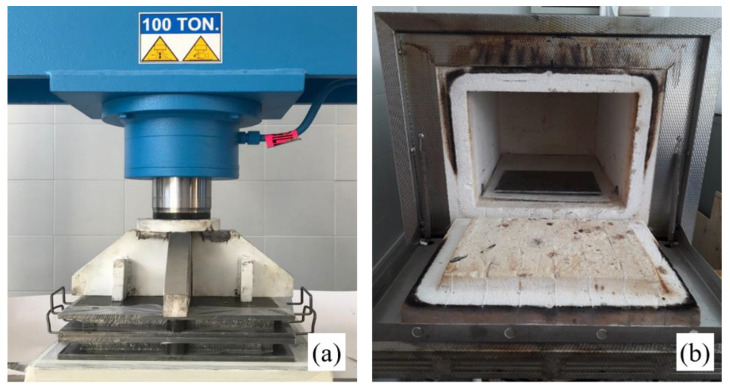
(**a**) Compression-molding performed by hydraulic press, and (**b**) post-curing performed by electrical oven.

**Figure 3 polymers-14-01457-f003:**
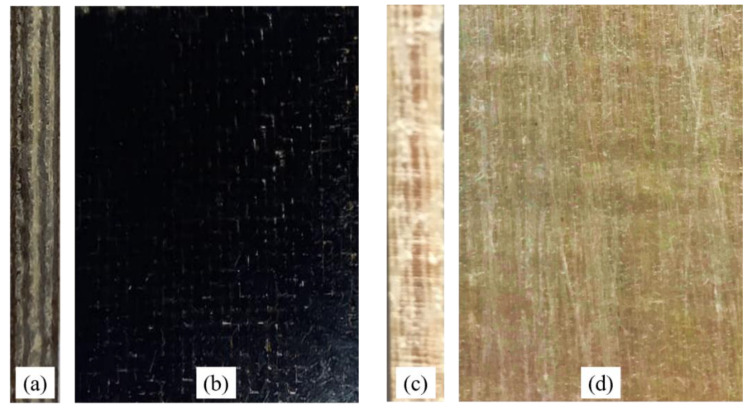
(**a**) Lateral view and (**b**) front view of the B-SFRP; (**c**) lateral view and (**d**) front view of the SFRP.

**Figure 4 polymers-14-01457-f004:**
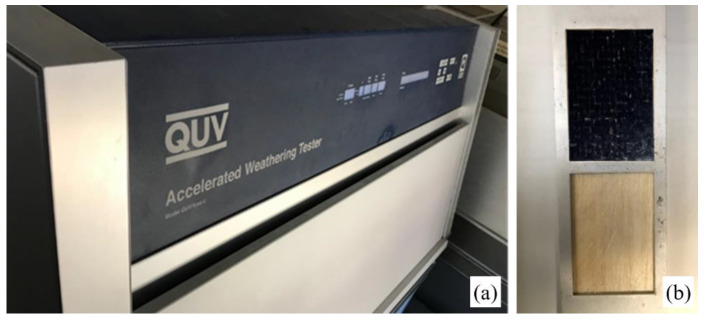
(**a**) QUV-accelerated weathering tester; (**b**) specimen rack placed into the aging chamber.

**Figure 5 polymers-14-01457-f005:**
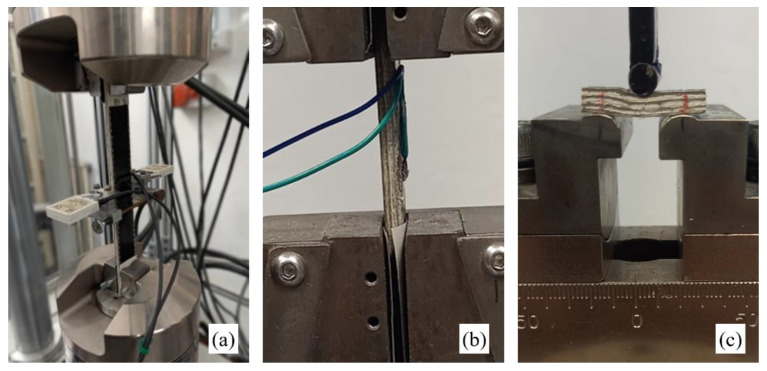
(**a**) Tensile test, (**b**) compression test, and (**c**) ILSS test.

**Figure 6 polymers-14-01457-f006:**
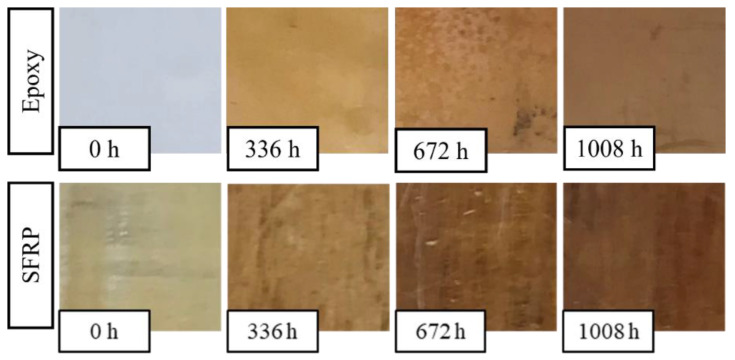
Epoxy and SFRP specimens subjected to different accelerated aging steps.

**Figure 7 polymers-14-01457-f007:**
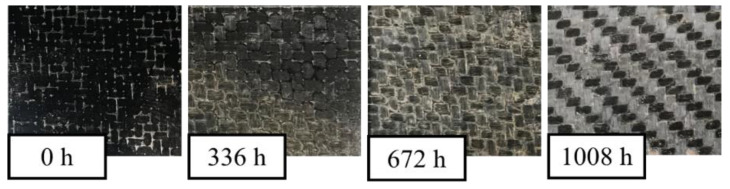
B-SFRP specimens subjected to different accelerated aging steps.

**Figure 8 polymers-14-01457-f008:**
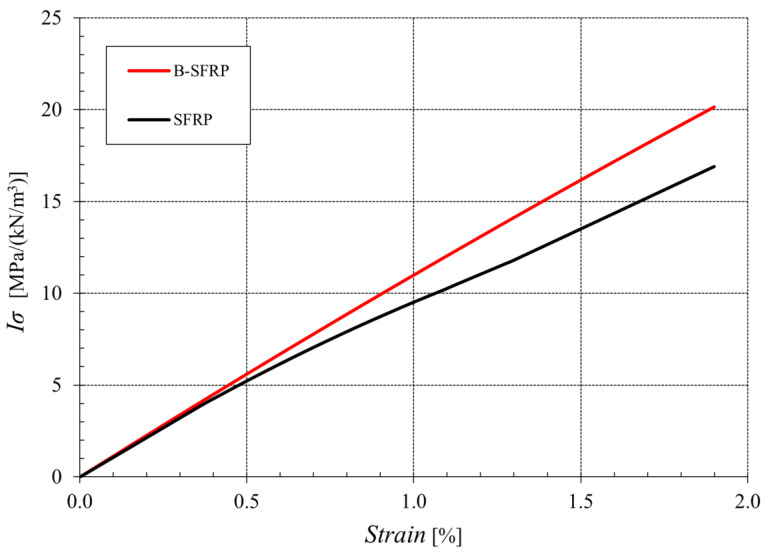
Specific tensile curves of the examined biocomposites SFRP and B-SFRP (without aging).

**Figure 9 polymers-14-01457-f009:**
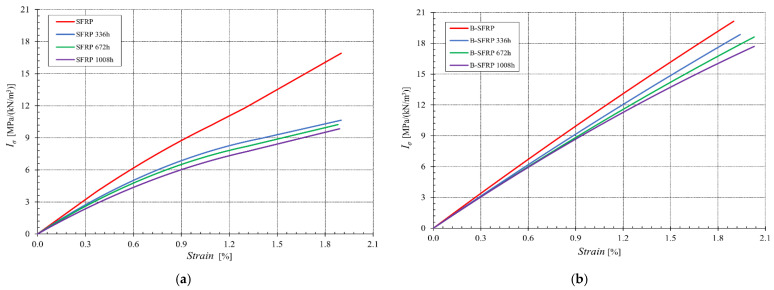
Tensile curves of (**a**) SFRP and (**b**) B-SFRP biocomposites before and after progressive aging.

**Figure 10 polymers-14-01457-f010:**
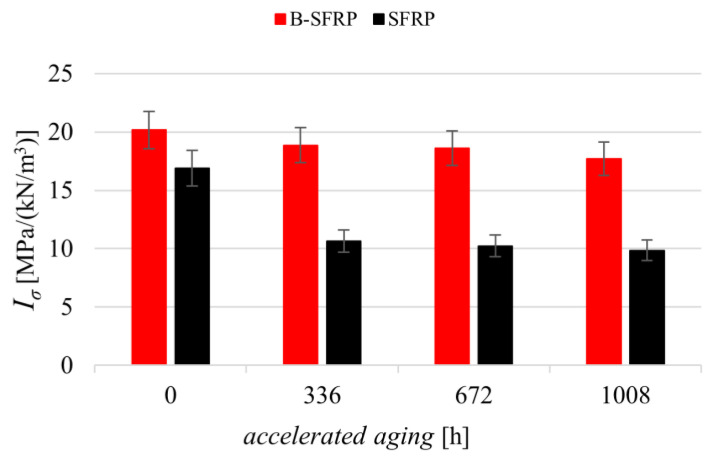
Specific strength of the SFRP and B-SFRP before and after the progressive aging.

**Figure 11 polymers-14-01457-f011:**
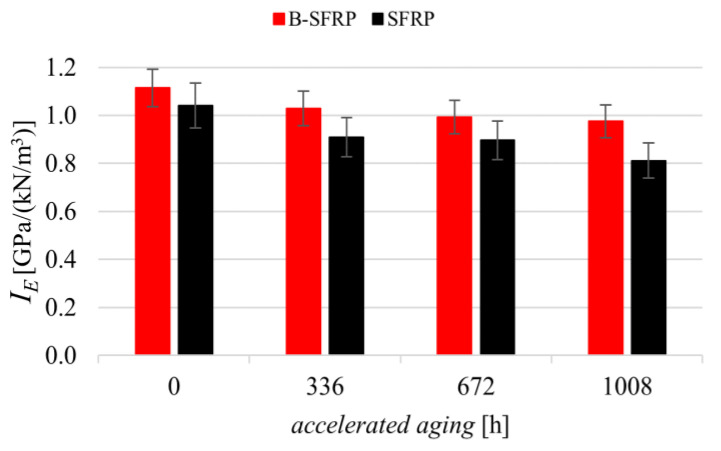
Specific stiffness of the SFRP and B-SFRP before and after the progressive aging.

**Figure 12 polymers-14-01457-f012:**
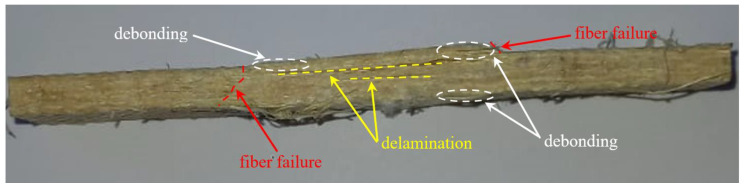
Typical SFRP specimen failure after tensile test.

**Figure 13 polymers-14-01457-f013:**
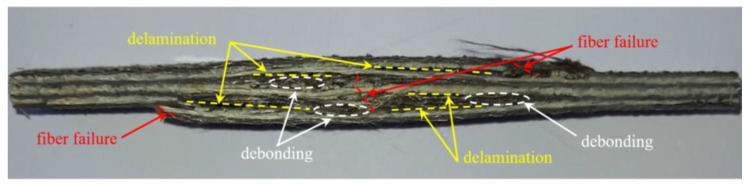
Typical B-SFRP specimen failure after tensile test.

**Figure 14 polymers-14-01457-f014:**
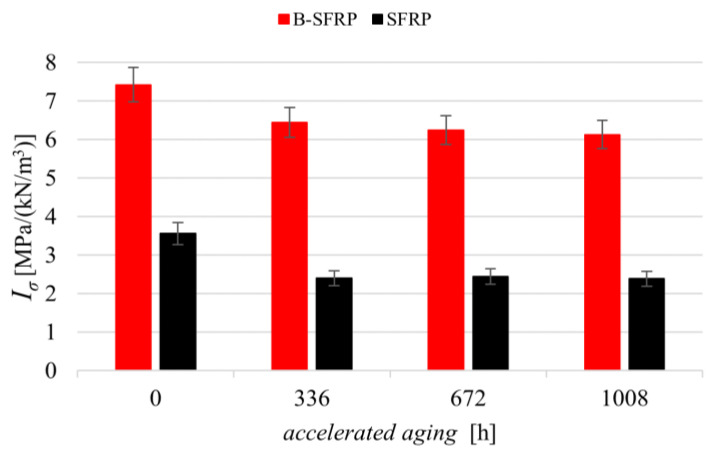
Specific compressive strength of SFRP and B-SFRP before and after progressive aging.

**Figure 15 polymers-14-01457-f015:**
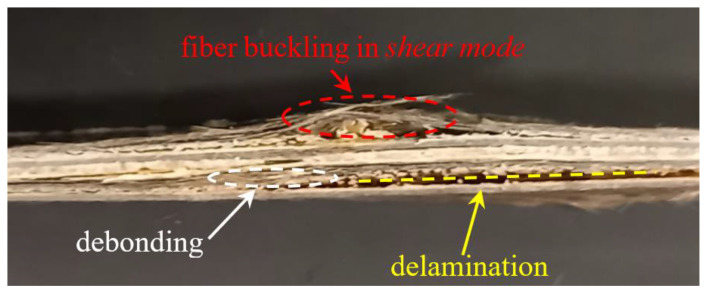
Typical damage of the B-SFRP specimen failure after compressive test.

**Figure 16 polymers-14-01457-f016:**
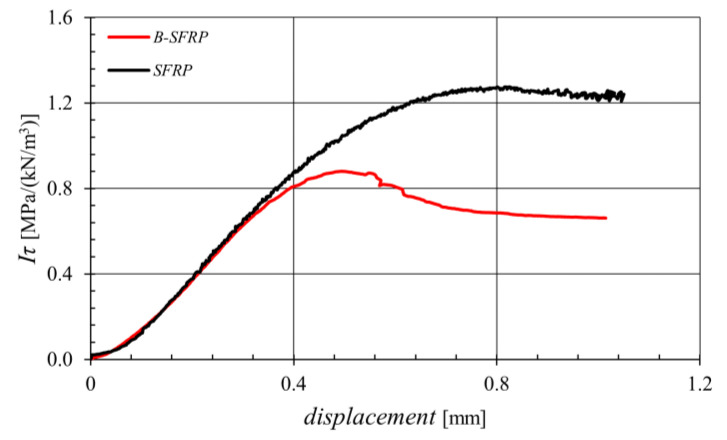
Specific shear curves obtained by the ILSS test for both the biocomposites examined.

**Figure 17 polymers-14-01457-f017:**
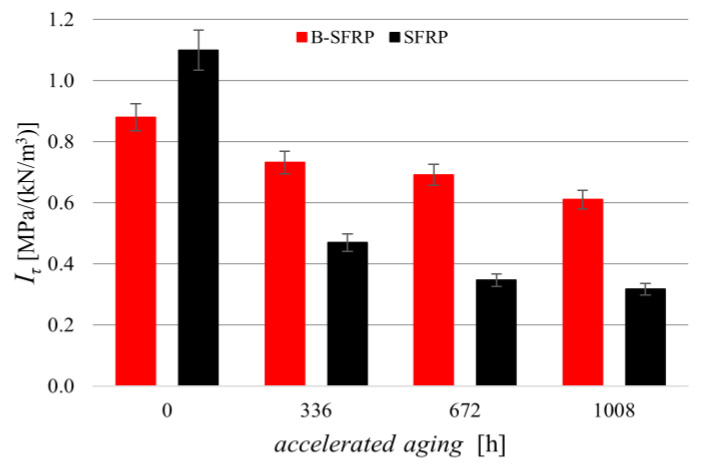
Delamination strength of the biocomposites examined, before and after accelerated aging.

**Figure 18 polymers-14-01457-f018:**
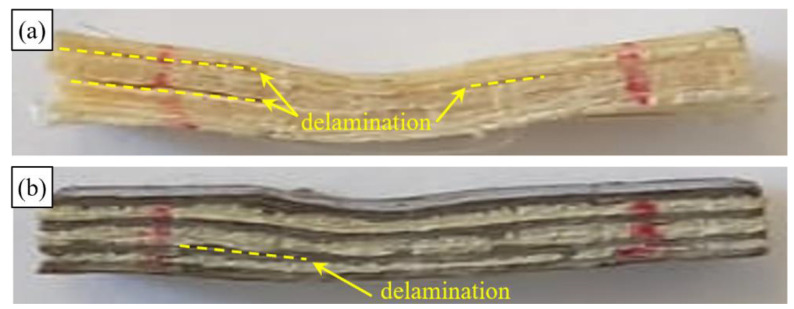
Typical failed ILSS specimen aged after 1008 h: (**a**) SFRP and (**b**) B-SFRP.

**Table 1 polymers-14-01457-t001:** Mechanical properties of materials used for sisal and basalt/sisal biocomposites.

Material	*ρ_f_* (g/cm^3^)	*σ_f_*_,*R*_ (MPa)	*E_f_* (GPa)	*ε_f_*_,*R*_ (%)
Sisal fibers	1.45	685	40	1.75
Basalt fibers	2.80	2800	89	3.15
Green epoxy	1.05	50	2.7	2.5

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
