# Peer review of "Basalt Fiber Hybridization Effects on High-Performance Sisal-Reinforced Biocomposites"

_polymers, 2022, doi:10.3390/polym14071457_

Round 1

Reviewer 1 Report

In my opinion it is a very interesting article about the properties of Basalt/Sisal reinforced hybrid Biocomposites. 

Before printing there are a few minor changes necessary:

  • page 1 / line 8: "Abstract: Abstract:" --> "Abstract:"
  • page 1 / lines 9, 10, 14, 23, 25 & 26: omit the unnecessary dash ("-" --> "")
  • page 4 / line 183, page 5 / lines 202 & 209: the unit kN/m³ is force per volume, but not really density. Please present the specific weight values as g/cm³ as used in Table 1.
  • page 15 / line 482: "Funding: Please add:" --> "Funding:"

Author Response

Responses to the Reviewers
First, the authors wish to thanks the Reviewers for their precious observations and suggestions that
have permitted to improve furtherly the quality of the manuscript and of the research work. The
text added in the revised version of the manuscript has been written in red, whereas the text
cancelled, has been eliminated definitively. In brief, the manuscript has been amended in
accordance with almost all requests, as described in the following:

Reviewer # 1

1) page 1 / lines 9, 10, 14, 23, 25 & 26: omit the unnecessary dash ("-" --> "")

R1: the double “Abstract” has been removed.

2) page 1 / lines 9, 10, 14, 23, 25 & 26: omit the unnecessary dash ("-" --> "")

R2: the unnecessary dash has been eliminated.

3) page 4 / line 183, page 5 / lines 202 & 209: the unit kN/m³ is force per volume, but not really
density. Please present the specific weight values as g/cm³ as used in Table 1.

R3: the correct unit density has been used.

4) page 15 / line 482: "Funding: Please add:" --> "Funding:"

R4: the erroneously words “Please add” has been eliminated.

Reviewer 2 Report

In this manuscript, Bernardo et al. investigated the effect of aging on properties by testing the mechanical properties of sisal/basalt hybrid biocomposites. The argument of the manuscript is more or less clear and organised. However, some modifications are required.

Line 8 contains two "Abstract"; please delete one.

The quality of the language, if the manuscript is not outstanding, still needs much improvement to be considered for publication.

The authors nicely presented some previous works (Lines 36-151) but could not clarify the gap in the existing literature.

Figure 6 is too blurry to justify with photographic detail "this darkening involves both the epoxy matrix and the sisal fibre." on line 278. Please replace it with a clear photo by the author.

Please rewrite the conclusion section briefly, and it is too long; it seems a new discussion has been started.

How many samples were tested? Consider adding statistical error in Figures 10, 11, 14 and 17. The authors might refer to https://doi.org/10.3390/ma12081226

To increase the quality of the presentation, I recommend authors add Figures 18 and 19 in one and mark them a and b.

Figure 4 should be deleted.

In place of Figure 15, the authors might see the interface using SEM.

The tensile properties are pretty ambiguous. According to my personal experience, the neat epoxy yields higher strength than these composites. What is the rationale for adding these fibres in it; it seems they have losen the mechanical properties by adding them.

Also, stress-strain curves are too good to be true.

References 29-32 should end with ".", not ";". They seem incomplete.

Do authors agree that similar other works might also predict these results in the literature related to composites; what the rationale for testing them is?

Author Response

R7: Figs. 18 and 19 have been reduced in a unique figure constituted by two sub-figures a)
and b).

8) Figure 4 should be deleted.

R8: Since Fig.4 depicts a device specialized in the accelerated test, carried out in a strict
accordance with the ASTM G154 standard (whereas most of the authors use home made
devices to implement similar tests approximately), the authors ask to keep this figure in the
manuscript.

9) In place of Figure 15, the authors might see the interface using SEM.

R9: In the opinion of the authors (that have used many times SEM for composite materials
analysis) the damage mechanisms, as delamination but also debonding and fiber buckling
that, as expected, occurs in the hybrid biocomposites examined in presence of compressive
loading, can be observed sufficiently by a direct visual analysis; in fact, sisal fibers have
relatively high diameters (about 0.2 mm vs 0.01 mm of the synthetical fibers) so that
debonding and buckling phenomena are visible at naked eye. Also delamination are clear
visible at naked eye since it involve very large portions of the specimens tested. The use of
SEM in general is more adequate for synthetical fibers or also to natural fibers to possible
observe splitting phenomena or local secondary pull-out, whereas in the opinion of the
authors, it does not allow to obtain further appreciable information in the examined case.

10) The tensile properties are pretty ambiguous. According to my personal experience, the neat
epoxy yields higher strength than these composites. What is the rationale for adding these
fibres in it; it seems they have losen the mechanical properties by adding them.

R10: The authors afraid that the observation of the Reviewer concerns the mechanical
strenght of short fiber biocomposites, in which the introduction of natural fibers allows to
increase significantly the stiffness of the polymer matrix, without none improvement of its
mechanical strenght. This is due to the fact that short fibers lead to appreciable stress
concentrations, without any exploitation of their longitudinal strenght, since a short fiber
biocomposite fails always through transversal failure mechanisms (with diffuse debonding,
fiber splitting, fiber pull-out etc. and rare phenomena of longitudinal tensile failure of the
fibers).

The high performance biocomposites examined in the present work are instead reinforced
by long fibers which allow not only to improve significantly the longitudinal stiffness, but also
to obtain noticeable mechanical strenght increases. Perhaps, the use of the specific tensile
strenght in Figs.8-10 has misled the Reviewer: the analyzed cross-ply biocomposite laminate
reinforced only by sisal fiber (SFRP) has a specific tensile strength of about 17 MPa/(kN/m3),
that corresponds to an absolute tensile strenght of 213 MPa that is very higher than the
strenght of the neat "green" epoxy used (equal to about 50 MPa as reported in the same
manuscript). Also, as it has shown in previous works of the same authors (see Refs.), if
unidirectional laminates of SFRP are considered, than the longitudinal tensile strenght
increases furtherly, up to values of about 470 MPa (one order of magnitude higher than that
of the neat epoxy resin).

11) Also, stress-strain curves are too good to be true.

R11: They are the curves acquired with the load cell of the test machine indicated in the
manuscript. Only the instable final part relative to the failure is not represented because the
test is interrupted as soon as the failure begin; this also to avoid the potential damaging of
the extensometer used to measure the longitudinal strain of the specimen.

12) References 29-32 should end with ".", not ";". They seem incomplete.

R12: In accordance with the observationof the Reviewer, the punctuation of Refs.29-32, has
been corrected.

13) Do authors agree that similare other works might also predict these results in the literature
related to composites; what the rationale for testing them is?

R13: Although in the end part of the Introduction section the authors have stated that the
research work reported in literature shown that in general the basalt hybridization of
composites leads to obtain appreciable mechanical strenght improvements, such a
statement is not equivalent, at all, to say that the present works is not useful and its results
could be predicted by similar works of literature. In fact, none previous work has never
considered the basalt hybridization of high performance biocomposites reinforced by long
sisal fibers (biocomposites for structural applications), but only glass hybridization of
biocomposite reinforced by short sisal fiber (the biocomposites for non-structural
applications, whose tensile strenght is lower than that of the neat matrix) and the effects of
hybridization on the ageing strenght has never studied.

Round 2

Reviewer 2 Report

It is considerable